# Exploring the distance between nitrogen and phosphorus limitation in mesotrophic surface waters using a sensitive bioassay

Enis Hrustić[1], Risto Lignell[2], Ulf Riebesell[3] and Tron Frede Thingstad[4]

[1]University of Dubrovnik, Institute for Marine and Coastal Research,

Kneza Damjana Jude 12, PO box 83, 20000 Dubrovnik, Croatia

[2]Marine Research Centre, Finnish Environment Institute, Mechelininkatu 34a,

PO Box 140, 00251 Helsinki, Finland

[3]GEOMAR Helmholtz Center for Ocean Research Kiel, Düsternbrooker Weg 20, D-24105 Kiel, Germany

[4]Department of Biology and Hjort Centre for Marine Ecosystem Dynamics,

University of Bergen, PO box 7803, 5020 Bergen, Norway

Correspondence to: T.F. Thingstad (frede.thingstad@uib.no)

## Abstract

The balance in microbial net consumption of nitrogen and phosphorus was investigated in samples collected in two mesotrophic coastal environments: the Baltic Sea (Tvärminne field station) and the North Sea (Espegrend field station). For this, we have refined a bioassay based on the response in alkaline phosphatase activity (APA) over a matrix of combinations in nitrogen and phosphorus additions. This assay not only provides information on which element (N or P) is the primary limiting nutrient, but also gives a quantitative estimate for the excess of the secondary limiting element ($P^+$ or $N^+$, respectively), as well as the ratio of balanced net consumption of added N and P over short time scales (days). As expected for a Baltic Sea late spring-early summer situation, the Tvärminne assays (n=5) indicated N-limitation with an average $P^+=0.30\pm0.10$ µM-P, when incubated for 4 days. For short incubations (1–2 days), the Espegrend assays indicated P-limitation, but the shape of the response surface changed with incubation time, resulting in a drift in parameter estimates toward N-limitation. Extrapolating back to zero incubation time gave P-limitation with $N^+\approx0.9$ µM-N. The N:P ratio (molar) of nutrient net consumption varied considerably between investigated locations; from $2.3\pm0.4$ in the Tvärminne samples to $13\pm5$ and $32\pm3$ in two samples from Espegrend. Our assays included samples from mesocosm acidification experiments, but statistically significant effects of ocean acidification were not found by this method.

Keywords: alkaline phosphatase activity, bioassays, mesotrophic temperate seas, nutrient limitation, phytoplankton

## 1 Introduction

N to P balance is a core biogeochemical feature of aquatic systems as highlighted in Redfield´s classical question of whether it is the chemistry of seawater that has determined the stoichiometry of the marine organisms, or biology is the cause for the "normal" 16:1 (molar) ratio between N and P in seawater (Redfield et al., 1963). The

issue of surface ocean nutrient limitation is as acute as ever (Moore et al., 2013), since it has bearings on phenomena ranging from the global carbon cycle, where it plays a key role in the dynamics of the ocean´s biological pump (Ducklow et al., 2001); via basin scale issues such as N deficiency in Arctic water of Pacific origin (Lehmann et al., 2005), P deficiency in Eastern Mediterranean deep waters (Krom et al., 1991) and the North Atlantic gyre (Mather et al., 2008); via regional issues such as the question of P and/or N removal from the Baltic Sea (Elmgren and Larsson, 2001; Granéli et al., 1990; Räike et al., 2003); to local ecosystem characteristics such as P-deficient brackish layer overlaying potentially more N-limited marine waters in the fjords of western Norway (Thingstad et al., 1993). The classical idea of predominantly N-limitation in marine systems (as opposed to predominantly P-limitation in limnic systems) (Hecky and Kilham, 1988) has also become considerably more nuanced, not only due to the cases mentioned above, but also with the identification of the High Nutrients Low Chlorophyll (HNLC) areas as being iron-limited (Franck et al., 2003), phosphorus and iron as co-limiting elements of nitrogen fixation in the tropical North Atlantic (Mills et al., 2004) and $N_2$ fixers potentially having a competitive advantage in oligotrophic P-starved regions (Landolfi et al., 2015). While some of the mechanisms behind these apparent deviations from Redfield stoichiometry seem to be well understood, there are others which lack generally accepted explanations.

In deep waters with most of the bioavailable N and P converted to $NO_3$ and $PO_4$, the chemical determination of N or P in excess of the Redfield ratio may be relatively straight forward. In biogeochemistry this excess is calculated on the basis of measured nitrate and phosphate, and is referred to as N* and P*, e.g. $N^*=NO_3−16PO_4+2.9$ mmol m$^{-3}$ (Sarmiento and Gruber, 2006). In productive surface waters this is a more complex issue. A potential solution to the chemically intractable problem of measuring a large suite of presumably bioavailable pools is to use a quantitative bioassay, i.e. to ask the organisms how much of the primary and secondary limiting elements they can "see".

In productive waters, both N and P may accumulate over time in pools of DON and DOP with different grades of bioavailability. Microbes have flexible stoichiometry as their content of storage materials, structural carbohydrates, nucleic acids and lipids vary with growth conditions (Bertilsson et al., 2003; Geider et al., 2002; Krauk et al., 2006). There are also differences in the stoichiometry of different functional groups of organisms, where e.g. bacteria (Fagerbakke et al., 1996) tend to have N:P ratio significantly lower than 16. Which element that first becomes limiting, and how much of the secondary limiting element then remains in excess may thus depend not only on the total pools as conceptually expressed by N*, but to vary as a function of the biological structure of the food web and its pre-history. Although conceptually related to N*,P*, the answer to what excess nutrients the organisms "see" may therefore differ even between systems with the same, chemically defined N*. We therefore have chosen to use the symbols $N^+$ and $P^+$ for surplus nitrogen and phosphorus as determined by bioassays, to distinguish these numbers from their chemically defined analogs N* and P*.

Microorganisms have evolved sophisticated physiological mechanisms to adapt to the different forms of nutrient limitation (Geider et al., 1997; Ivančić et al., 2012; Thingstad et al., 2005; Van Mooy et al., 2009; Lin et al., 2016), including the induction of extracellular enzymes such as alkaline phosphatase (AP) catalyzing the hydrolysis of phosphomonoesters within DOP (Hoppe, 2003). A well-studied model system is the induction of the Pho-regulon in *Escherichia coli*, which leads to expression of a series of P-starvation related genes, including *phoA* coding for AP synthesis (Torriani-Gorini, 1994). The induction of AP synthesis seems to be more coupled to a low internal cell

quota of P, than directly to low external concentrations of inorganic P (Lin et al., 2016), thus presumably providing a main signal when both external pools and internal storage reserves of P have been depleted below the certain level (Boekel and Veldhuis, 1990; Chróst and Overbeck, 1987). Inducible AP synthesis is a wide-spread feature in microorganisms (Jansson et al., 1988). It is easily measured as AP activity (APA) (Perry, 1972, Hoppe, 2003), and thus it has been frequently used as an indicator of P-stress (Jansson et al., 1988; Dyhrman and Ruttenberg, 2006; Lomas et al., 2010).

This method was further exploited by Thingstad and Mantoura (2005) in the oligotrophic Eastern Mediterranean, showing that the concentration of added $PO_4$ needed for APA to disappear in a P-limited system, or alternatively the $NH_4$ needed to induce APA in an N-limited system, could be used as a bioassay to quantitatively estimate $N^+$ and $P^+$, respectively. We here expand this technique by using a matrix-setup including simultaneous gradients in both $PO_4$ and $NH_4$ additions. This is applied to samples from the coastal waters of western Norway and the Baltic Sea, confirming that the assay gives informative results also in temperate, mesotrophic environments.

## 2 Material and Methods

2.1 Study areas and sampling

Part of the sampling for this study was performed in mesocosms designed to study acidification effects. In the Baltic, the water was collected as integrated samples (depth 0–10 m) in Storfjärden near Tvärminne field station (59$^o$ 51.50' N, 23$^o$ 15.50' E) on 6 August 2012. The collection was performed 45-30 days after the first-last $CO_2$ treatments and 50 days after the mesocosm closure (Paul et al., 2015). Samples were collected from the fjord (417 µatm) and mesocosms M1 (365 µatm), M3 (1007 µatm), M6 (821 µatm) and M7 (497 µatm); where numbers in parentheses are average f($CO_2$) over the period Day 1–Day 43. The mesocosms received no nutrient manipulations except the $CO_2$ treatments. Further details about location and the mesocosm experiment can be found in Paul et al. (2015) and Nausch et al. (2016).

The samples from western Norway were collected during a similar mesocosm experiment in Raunefjorden close to Espegrend field station (60$^o$ 16.2' N, 5$^o$ 11.7' E). From one mesocosm (MR) an integrated (depth 0–20 m) sample (1165 µatm) was collected on 25 May 2015 corresponding to Day 22 after acidification treatment. The fjord sample was collected at nearby landlocked location Kviturspollen (60$^o$ 15.8' N, 5$^o$ 15' E) at the depth of 1 m using a Niskin sampler on 3 June 2015. Samples were pre-filtered through gauze of 112 µm mesh size to minimize the variability due to the occasional large zooplankton.

2.2 Matrices of nitrogen and phosphorus additions

Samples were distributed in 15 mL Falcon® polypropylene tubes (BD Biosciences®) organized in 10x10 or 8x8 columns x rows (Tvärminne and Espegrend, respectively). $PO_4$ (KH$_2$PO$_4$ 10 µM) was added in final concentrations from 0 to 290 nM-P in steps of 32.2 nM (Tvärminne) and from 0 to 105 nM-P in steps of 15 nM (Espegrend). Each of the columns received additions of $NH_4$ (NH$_4$Cl 200 µM) in final concentrations from 0 to 964 nM-N in steps of 107 nM-N (Tvärminne) and from 0 to 2100 nM-N in steps of 300 nM-N (Espegrend). The tubes were incubated in light:dark (16 h:8 h) at 17–18°C (Tvärminne) and in light:dark (12 h:12 h) at 16.5$^o$C (Espegrend), both at irradiance of 78 µmol

photons $m^{-2}$ $s^{-1}$. Incubation at Tvärminne lasted 4 days for all samples, whilst APA assays
for Espegrend were repeated as given in each case.

2.3 Alkaline phosphatase activity

Measurements of APA were done according to Perry (1972) using 3-o-methyl-
fluorescein-$PO_4$ (final concentration 0.1 μM) as the substrate. Volumes were modified to
the use of fluorescence plate reader by pipetting 200 μL subsamples from each Falcon
tube into the wells containing the substrate. Results are expressed as increase in relative
fluorescence units per hour (RFU $h^{-1}$). APA in the coastal waters of the western Norway
was measured using a PerkinElmer Enspire 2300 plate reader programmed to do 15
repeated measurements (time interval 5 min) over a total incubation time of 70
minutes. APA was calculated as the slope of the fitted linear regression line. APA in the
Baltic Sea was measured by Varian Cary Eclipse fluorometer after 30 minutes incubation
with substrate.

2.4 Fitting the response surface

To interpret the data obtained by this method, an objective algorithm is needed to define
the transition between subsamples with high (P-limited) and low (N-limited) post-
incubation APA. Thingstad and Mantoura (2005) did this by fitting sigmoidal functions
to the observed APA-responses; either a decreasing function parallel to the P-addition
axis in the case of a P-limited system, or an increasing function parallel to the N-addition
axis in the case of N-limitation. To avoid this pre-fitting choice of function, we here have
instead started with the assumption that the $P,N$-plane is split into a P-limited and an N-
limited region by the straight line:
$$N = N_0 + rP \quad \text{Eqn. 1}$$
where a negative value of the intercept $N_0$ corresponds to the excess-N ($N^+$) present in a
P-limited system and $P_0 = \frac{-N_0}{r}$ is the amount of phosphate needed to shift the system to
N-limitation. Conversely, a positive value of the intercept $N_0$ would correspond to the
amount of N required to shift an N-limited system into P-limitation, while $P_0 = \frac{N_0}{r}$ then
corresponds to the excess-P ($P^+$) in this N-limited system. The shift from P- to N-
limitation, and therefore the expression of APA in a point $P,N$ is assumed to be a function
of the distance $Z$ between this point and the line (Fig. 1). The sigmoidal function fitted is:
$$APA_{est} = \frac{A}{1+e^{sZ}} - B \quad \text{Eqn. 2}$$
From the geometry of Fig. 1 one can calculate the perpendicular distance Z from the
point $P,N$ to the line defined by Eqn.1 as
$$Z(P,N) = \frac{1}{\sqrt{1+r^2}}(rP - (N - N_0)). \quad \text{Eqn. 3}$$

Here, the exponential function in the denominator of Eqn. 2 replaces the term $\left(\frac{Z}{Z_0}\right)^s$
adopted by Thingstad and Mantoura (2005) from standard calculation of lethal
concentration (i.e. $LC_{50}$) in toxicology. This standard expression is undefined for $Z_0=0$
and therefore not applicable with our approach where $Z = 0$ along the line defined by
Eqn. 1. Visual inspection of residuals in graphs (see Fig. 2A, B) did not suggest
systematic deviances between response surfaces fitted with this function and the
observed data. Alternative fitting functions have therefore not been explored. With five

parameters to fit ($r$, $N_0$, $s$, $A$, $B$), this leaves 95 and 59 degrees of freedom for the
Tvärminne and Espegrend set-ups, respectively. The fitted surface $APA_{est}$ has a
maximum $A$-$B$ obtained for co-ordinates combining low $P$ with high $N$ (large negative $Z$)
and $APA_{est}=(A/2)$-B along the line $N=N_0+rP$ separating the P- and N-limited regions. The
parameter $s$ defines the steepness of transition between the two regions perpendicular
to this line. $B$ is the background $APA_{est}$ found for high-$P$, low-$N$ (large positive $Z$) co-
ordinates. The fitting was done using the "fit" function in Matlab® with its default
Levenberg-Marquardt algorithm providing the parameter estimates with 95%
confidence intervals (c.i.) (code included in SI).

**3 Results**

Two examples of the fitted response surface, one from Tvärminne (Fjord) (Fig. 2A) and
one from Espegrend (MR) (Fig. 2B) are shown to illustrate the difference in shape of the
response in situations apparently N-limited (Tvärminne) and P-limited (Espegrend),
with estimated $P^+$=0.3 µM-P and $N^+$=0.4 µM-N, respectively . All assays are summarized
in Table 1.
For the two Espegrend samples, the change in shape of the response surface with
incubation time was explored (Fig. 3). For both samples, $N_0$ increased with incubation
time (p≤0.05, Table 2), i.e. the assay results drifted towards increasing N deficiency
when using longer incubation times. In the sample MR, $r$ and $s$ decreased significantly
over time (Table 2). Using linear regression, the parameter estimates can be
extrapolated back to zero incubation time. With this technique the average $P^+$ for the
Tvärminne samples, based on a single incubation time, was 0.3 µM-P, and the average $N^+$
for the two Espegrend samples, based on backward extrapolation, was 0.9 µM-N.
The assays from Tvärminne mesocosms include an $f(CO_2)$ gradient. Linear regressions of
$N_0$ (p=0.55), $r$ (p=0.63) (Fig. 4) and $s$ (p=0.19) (not presented) on $f(CO_2)$ gave no
indication of any statistically significant effect of the 45 days exposure of the systems to
different $CO_2$-levels. Compared to a Redfield N:P value of 16, all the Tvärminne samples
gave low $r$ (2.3±0.5; mean over samples±sd), while the two Espegrend samples gave $r$ of
13±2 (Kviturspollen) and 32±3 (MR) (mean±sd, both over incubation times).

**4 Discussion**

This study extends the demonstrated applicability of this type of assay from its previous
use in warm oligotrophic waters (Thingstad and Mantoura, 2005) to mesotrophic
temperate environments. We modified the technique so that no *a priori* assumptions are
now required as to whether the system investigated is N- or P-deficient. Note that the
function used to fit the response (Eqn. 2) was not derived from explicit assumptions on
biological mechanisms producing the response, but as a convenient statistical model
that fitted the observed responses without obvious systematic patterns in the residuals
(Fig. 2). It may, however, be of biological relevance to observe that, with this description,
the initially three-dimensional description ($P$, $N$, $APA$) is reduced to two dimensions ($Z$,
$APA$): all combinations of $P$ and $N$ that have the same perpendicular distance $Z$ (Eqn. 3)
to the line representing N:P balance (Eqn. 1) develop the same APA (Eqn. 2). Contour
plot representations of the fitted surfaces in Fig. 2 A and B would thus consist of straight
lines parallel to the line described by Eqn. 1.

We explored the use of this modified assay in two environments with anticipated differences in ambient N:P stoichiometry. The Tvärminne mesocosm experiment was planned with the expectation of an N-limited spring-summer situation as characteristic in the Baltic Sea (Granéli et al., 1990; Rolff and Elfwing, 2015; Thomas et al., 2003), subsequently transiting from N-limitation towards an N- and P-co-limited situation as the result of "new" N being added through late summer blooming of diazotrophic cyanobacteria (Lignell et al., 2003). This bloom did not occur during the whole Tvärminne experiment and N-limitation at the time of sampling has been confirmed by Nausch et al. (2016) who studied the microbial P-cycle just before our experiment. Nutrient concentrations were not significantly changing throughout the whole acidification experiment (Paul et al., 2015). DIN and DIP equalled ~0.25 µmol $L^{-1}$ and ~0.15 µmol $L^{-1}$, respectively, giving a ratio of 1.67 (Paul et al., 2015). Compared to a Redfield ratio of 16, these chemical determinations suggests N-deficiency; although not taking other bioavailable forms into account. Our finding of positive $N_0$-estimates for all 5 samples (Table 1) is in line with this. The Tvärminne assays were performed after the 4 days of incubation needed for the APA-responses to emerge. The conclusion of N-limitation is therefore confounded by the potential drift in parameter estimates as was later observed for the Espegrend samples (Fig. 3). The drift obviously complicates the use of this assay since there may be no single incubation time that gives a "correct" set of parameter values Since the drift seems to be reasonably linear for all parameters (Fig. 3), we see it as a promising option to extrapolate the linear regressions back to time 0, assuming this to give values representative for the initial conditions in the water sample. In our case this gives negative $N_0$ values of -0.8 (-1.4,-0.2) and -1.0 (-2.4,0.4) for the Espegrend samples from MR and Kviturspollen, respectively (intercept with 95% c.i.); suggesting initial P-limitation. This conclusion is in accordance with our expectation since the top layer of the fjords in western Norway has been shown to be P-deficient (Thingstad et al., 1993).

The mechanisms behind the drift in parameter estimates have not been studied further here. Three, not mutually exclusive, scenarios may, however, illustrate some of the theoretical possibilities: 1) The microbial food web in the incubated tubes remineralizes P faster than N (Garber, 1984). The assay may then correctly reflect the succession of the limiting nutrient in the sense that the bioavailable pools in the tubes change over time as N becomes immobilized in slowly degradable detritus to a larger extent than P. 2) N added in excess of P in the upper P-limited part of the *P,N*-plane is used by the organisms to produce alkaline phosphatase (rather than biomass). This would lift the response surface for high values of added N which may move the fitted line towards higher $N_0$, i.e. towards N-limitation. The use of extra N to produce exo-enzymes for acquisition of P from DOP has recently been argued for, but then with $N_2$-fixation as the N source (Landolfi et al., 2015). 3) Successions in the microbial food web move towards organism groups that require more N relative to P, although an increasing dominance of P-rich bacteria (Fagerbakke et al., 1966) would in this scenario produce a drift in the direction opposite to that observed. The *r* values representing the ratio of N- and P-net consumption are comparable between all the Tvärminne samples (2.3±0.5, n=5 different samples), indicating good reproducibility of the assay for similar water samples. This low value compared to the Redfield value of 16 was, however, strikingly different from the Espegrend samples with one Redfield-like 13±2 (Kviturspollen) and one high 32±3 (MR) value, both averaged over incubation times. A similar phenomenon was noted by Thingstad and Mantoura (2005) using this method to study in-out differences in a Lagrangian experiment where orthophosphate was added to the P-deficient surface

system in the Eastern Mediterranean. While their P-limited out-sample gave an $r=15\pm2$, the inside system, when driven to N deficiency by the *in situ* phosphate addition, gave a much lower $r=3.0\pm0.2$. Interestingly, we also here found the lower-than-Redfield $r$ values in the probably N-limited samples from Tvärminne. From the limited number of assays available, the linkage between N deficiency and low $r$ values thus seems consistent. In microorganisms, C:P-ratios are usually more flexible than C:N-ratios (Gismervik et al., 1996; Fagerbakke et al., 1996). P-rich microorganisms in N-deficient environments may thus seem a potential explanation to the observed low $r$-values in N-limited situations.

Considering the difference in sigmoidity ($s$) for the MR and Kviturspollen samples (Fig. 3) it seems that $s$ represents a characteristic of the initial water sample. While $s$ reflects the stoichiometric flexibility in the community response, it would require further investigations to determine whether this flexibility is at cell level and would be seen also in axenic cultures, or is a reflection of differences between species present.

## 5 Conclusions

We have demonstrated the extension of the APA assay from its previous use in warm oligotrophic, to temperate mesotrophic surface waters. The primary advantage of this technique over traditional nutrient-limitation bioassays is that it indicates which of the elements N or P is the most limiting, while simultaneously providing estimates of the excess in bioavailable forms of the secondary limiting element ($N^+$, $P^+$) along with the ratio between net consumption of the two elements ($r$). The assay does not require determinations of the large variety of chemical and/or physical forms in which the primary and secondary limiting elements may exist. The assay was found to be complicated by a drift in parameter estimates with incubation time. A backward extrapolation to zero incubation time appears promising. Further work is needed to fully understand the processes creating this drift and also the mechanisms that in some cases generate large deviations in $r$ from the Redfield value of 16. The consortium of ecological processes that create the APA response during incubation are likely to be relevant to processes shaping nutrient limitation in natural aquatic systems,. The experimental setup used in this assay thus seems also to have a potential as a tool for future studies on the ecological stoichiometry of aquatic microbial food webs.

## Data availability

Original data are given in Supplementary Information (SI) for each assay in the form of a Matlab$^{®}$ program that will also fit the response surface as shown in Fig. 2.

## Acknowledgements

The cooperation between Enis Hrustić and Professor Tron Frede Thingstad was realized through the Erasmus+ training at University of Bergen, Norway and EU project OCEAN-CERTAIN FP7-ENV-2013.6.1-1. Project nr. 603773. The mesocosm studies in Tvärminne and Espegrend were funded by BMBF projects SOPRAN Phase II (FKZ 03F0611) and

BIOACID II (FKZ 03F06550). MSc Johanna Oja is thanked for laboratory assistance in the Tvärminne experiment.

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

Table 1: Estimates (with 95% c.i.) of the intercept ($N_0$) and the slope ($r$) of the line
$N=N_0+rP$ separating the N- and P-limited regions as illustrated in Fig. 1. $s$ represents the
steepness of transition from N- to P-limitation, perpendicular to the line. $R^2$-values are
for the fitted response surfaces.

| | $N_0$ (µM-N) | $r$ (µM-N:µM-P) | $s$ (µM$^{-1}$) | $R^2$ |
|---|---|---|---|---|
| **Tvärminne (Baltic Sea): (incubation time 4 days)** | | | | |
| Fjord | 0.79 (0.75, 0.83) | 2.5 (2.2, 2.8) | 29 (23, 35) | 0.674 |
| M1 | 0.34 (0.30, 0.38) | 2.9 (2.7, 3.2) | 43 (31, 56) | 0.622 |
| M3 | 0.76 (0.69, 0.83) | 2.2 (1.9, 2.5) | 19 (15, 24) | 0.664 |
| M6 | 0.64 (0.56, 0.72) | 2.2 (1.9, 2.6) | 19 (13, 25) | 0.569 |
| M7 | 0.70 (0.66, 0.75) | 1.6 (1.4, 1.8) | 20 (16, 25) | 0.635 |
| **Mesocosm Raunefjorden** | | | | |
| Incubation time (days): | | | | |
| 1 | -0.41 (-0.60, -0.22) | 40 (36, 44) | 64 (52, 76) | 0.965 |
| 2 | -0.63 (-1.02, -0.25) | 34 (28, 40) | 50 (31, 70) | 0.764 |
| 3 | -0.21 (-0.47, 0.06) | 31 (27, 36) | 48 (33, 63) | 0.843 |
| 4 | -0.13 (-0.34, 0.08) | 31 (27, 34) | 47 (35, 59) | 0.933 |
| 4.5 | -0.13 (-0.30, 0.04) | 30 (27, 33) | 51 (40, 63) | 0.951 |
| 5 | 0.42 (0.15, 0.69) | 26 (23, 29) | 36 (25, 47) | 0.907 |
| **Kviturspollen** | | | | |
| 2 | -0.09 (-0.26, 0.08) | 9.9 (8.5, 11.4) | 25 (17, 33) | 0.940 |
| 3 | 0.58 (0.46, 0.70) | 12 (11, 13) | 20 (15, 25) | 0.972 |
| 4 | 1.83 (0.72, 2.95) | 14 (13, 16) | 15 (8, 22) | 0.946 |
| 5 | 1.65 (1.32, 1.99) | 15 (14, 17) | 25 (20, 30) | 0.963 |
| 7 | 2.84 (0.10, 5.59) | 14 (11, 16) | 18 (9, 27) | 0.855 |

1    Table 2. Linear regressions of parameter estimates against incubation time for the
2    Espegrend samples (see Fig. 3). Extrapolation to zero time is given (Day 0).

|  | Intercept (Day 0) | Slope of linear regression | p ($H_0$: slope$\neq$0) | $R^2$ |
|---|---|---|---|---|
| **MR** | | | | |
| $N_0$ | -0.8  (-1.4,-0.2) | 0.19  (0.01,0.37) | 0.05 | 0.674 |
| $r$ | 41  (36,46) | -2.8  (-4.1,-1.5) | 0.004 | 0.896 |
| $s$ | 65  (48,81) | -4.7  (-9.4,0.0) | 0.05 | 0.656 |
| **Kviturspollen** | | | | |
| $N_0$ | -1.0  (-2.4,0.4) | 0.6  (0.3,0.9) | 0.01 | 0.917 |
| $r$ | 10  (3,16) | 0.8  (-0.7,2.3) | *0.2* | 0.479 |
| $s$ | 24  (6,42) | -0.8  (-4.7,3.1) | *0.6* | 0.121 |

3    $H_0$  null hypothesis

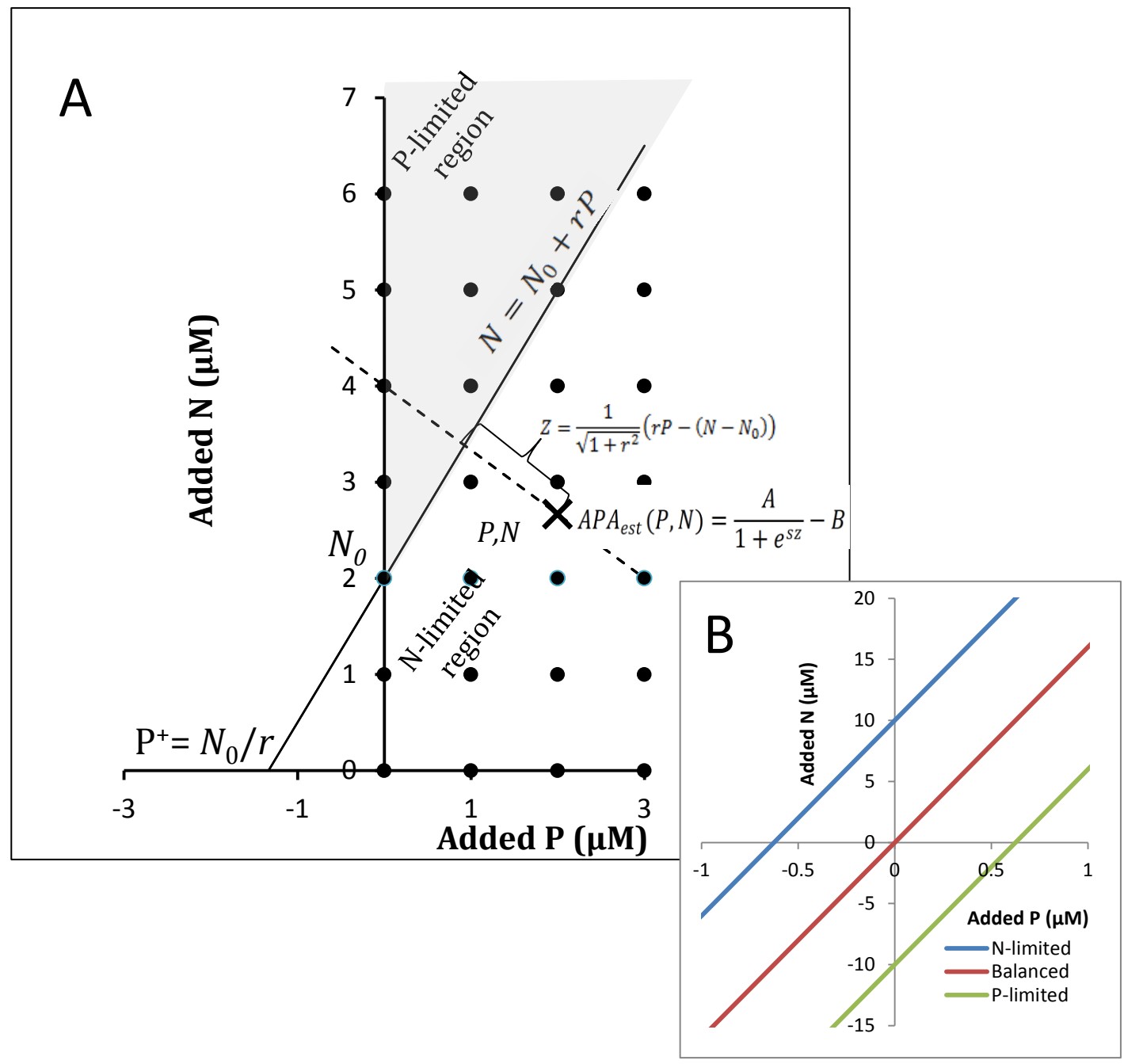

Figure 1. Illustration of the fitting algorithm used. With APA measured over a 4x7 matrix of
combinations in additions of P and N (black dots, Panel A), the objective is to find the line that splits
this *P,N*-plane in an upper P-limited region with high APA (shaded, Panel A) and a lower N-limited
region with low APA. This is done by least square fitting of the surface $APA_{est} = \frac{A}{1+e^{sZ}} - B$ to the
APA-values measured in each grid point. $APA_{est}$ is a sigmoidal function of the perpendicular
distance $Z = \frac{1}{\sqrt{1+r^2}}(rP + (N - N_0))$ from the point *P,N* (marked X in Panel A) to the line. The
situation illustrated in Panel A represents an N-limited system with the positive *N*-axis intercept
($N_0$) and excess-P (P+) represented by the negative *P*-axis intercept $N_0/r$. Panel B illustrates the
separating line for three hypothetical situations characterized by N-limitation (blue), N/P balance
(red) and P-limitation (green). All three with r = 16, but with P+ = 0.625 μM-P, P+ = N+ = 0, and N== 
10 μM-N for the N-limited, the balanced, and the P-limited situation, respectively.

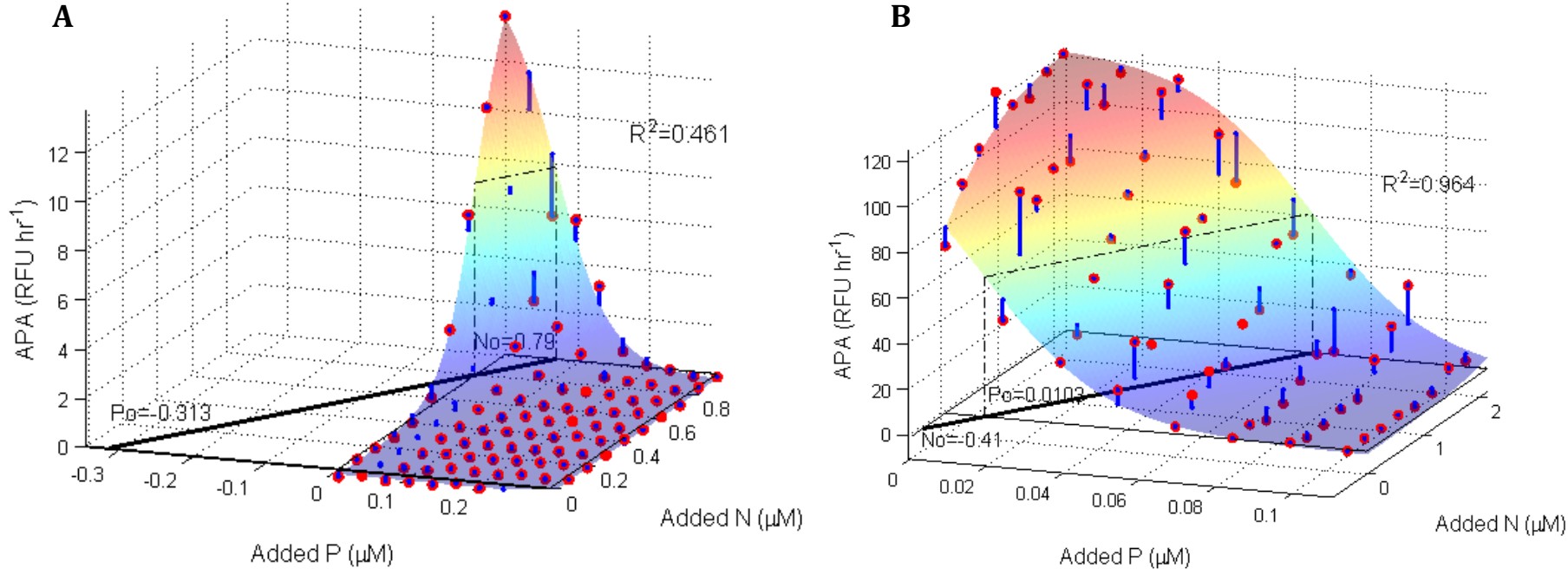

3 Figure 2. Measured APA values (red), fitted response surface (blue – red gradient from N- to P-limited) and residuals (blue) for assays:
4 (A) Fjord from Tvärminne and (B) MR from Espegrend (Day 1); illustrating situations interpreted as N-limited with $P^+=0.3$ μM-P and a P-
5 limited with $N^+=0.4$ μM-N, respectively.

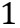

3    Figure 3. Change in parameter estimates with incubation time for the two samples from western
4    Norway. Kviturspollen has filled symbols, mesocosm MR has open symbols.

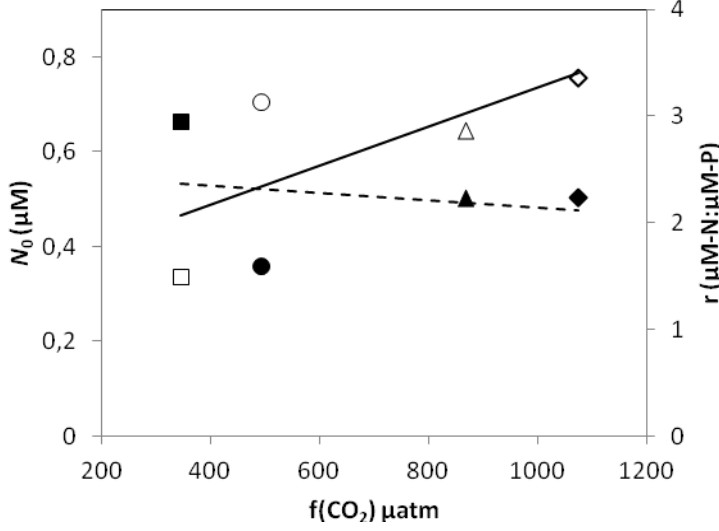

2  Figure 4. Scatterplots between f($CO_2$) and estimates of $N_0$ (open symbols, solid regression line) and
3  $r$ (closed symbols, dotted regression line) for Tvärminne mesocosms M1 (squares), M3 (diamonds),
4  M6 (triangles) and M7 (circles). Regression slopes are not significant (p=0.27 and 0.79,
5  respectively).