# Peer review of "limitation in mesotrophic surface waters using a sensitive"

_Biogeosciences, 2016_

## Short Comment (SC1) · 2 Aug 2016

I have read the manuscript in detail. I was peripherally involved in the Thingstad and Mantoura (2005) study and found it interesting and useful. This manuscript develops those ideas further and generalizes them to quantify the degree that a system is N or P limited. As such this work fills an important gap in the literature. I am happy to recommend publication and cannot recommend any changes to the manuscript as written. However I would be less than honest if I did not say that I got lost in the detail of exactly how the results are plotted and then interpreted. As far as I can see this is done well but I would defer to another reviewer if (s)he were more expertise in the mathematics of such biological measurements and finds either that the text does not

explain this part well enough or even that there were problems in the calculations as done. I simply don't have the expertise to know.

---

## Referee Comment (RC2) · Anonymous Referee #2 · 27 Sep 2016

The authors present a data set and analysis method aimed at diagnosing and quantifying the excess bioavailabilities of N and P within aquatic systems. The new data presented appears to be a rich resource and the proposed analysis technique appears to have merit. Having said this, I found the manuscript difficult to assess on first reading and I would encourage the authors to consider expanding both the text and presentation of the data to make accessibility clearer to the general reader. Below I make some general comments and suggestions which I would like to see the authors address in any revised version of the manuscript.

Specific comments:

Both conceptually and in terms of practical calculation the estimates of 'N*' and 'P*'

derived from the experimental protocol described differ from the typical usage, which has in the past been based on a strict definition (e.g. N* = Nitrate – 16 x Phosphate). In order to avoid confusion within the literature I would prefer the use of different terminology (e.g. maybe 'apparent excess N or P') or, at the very least, a strict definition of the terms used. In particular, at present I remain unsure how and whether the resulting values of derived 'N*' and 'P*' might depend on system characteristics, e.g. including the biological characteristics of the system. Indeed, given the time dependence of the results, it seems likely that derived values are a complex and variable system property. Additionally, I would like to see the authors' thoughts on whether the variable sizes and turnover times of bioavailable dissolved organic N and P pools might influence their results? Additionally, I was unclear how the presence and turnover of natural DOP pools, both AP-hydrolysable and otherwise might be influencing the results?

Related to the above, it would be useful to have some additional information on the systems studied, e.g. including inorganic and organic N and P concentrations and potentially some characteristics of the planktonic community?

Given the focus of the paper, I would like to see some more details on the APA measurements. In particular what substrate was used (MUFF-P?) and at what concentration. Also how long were the APA incubations and were multiple timepoints measured within the incubation? Concentrations of substrate are potentially particularly important and I would be interested in the authors' opinion on whether it would matter if the substrate concentration was saturating or not for the derived response surface?

Some of the measured APA values are negative, what does this represent? I assume some form of blank correction within the fluorometric measurements? Please be more specific on methods.

Additional comments on specific sections within the text:

Abstract:

Line 19: I suggest '. . .primary limiting nutrient. . .'

Section 1:

Line 16: suggest '. . .fixers potentially having a competitive. . .' i.e. the evidence in the cited study is largely on the basis of hypothetical modelling

Line 22: define N* and P* (e.g. N* = Nitrate – 16 x Phosphate) on first use (also note above, I would prefer use of different terminology for derived apparent excesses of N and P)

Line 44: It was not immediately clear to me that the assays described can be used to quantitatively asses the excess of overall bioavailable N or P in a system and, despite spending some time with the authors' code and data (which are very usefully provided within the supplement), I still remain to be entirely convinced. Firstly, there is clearly an assumption that the organisms in the community react to P limitation through increasing APA (see abstract of Thingstad and Mantoura 2005). While this may be the case, it remains an assumption and should probably be explicitly stated as such. I think there is also an underlying assumption that the community level APA response is directly proportional to the overall availability of all forms of bioavailable N and P within the system. The authors can correct me if this is wrong, however, I think that the equation on line 2, page 4 suggests there is this assumption of a linearity of response? Overall however I will admit that I struggled to follow some of the authors' arguments, particularly on the first read through. I would therefore suggest that the authors could provide a fuller treatment and explanation for their analysis method for the data presented.

Related, it would be good to fully separate out the equations, number them and explicitly refer to them as/when required. Some additional simple graphical schematic plots and further graphical presentation of the extensive data set might also aid accessibility (see below).

Section 2:

Line 4: Rephrase. Please provide some background to the experiment. At this stage the experimental mesocosms have not been described/introduced.

Line 30 (section 2.3): please provide information on both the incubation time for APA, substrate used, concentration etc. (see above).

Page 6, lines 1-12: given that the value of r has a direct impact on the calculations of any excess bioavailable N or P within the system, I was left wondering how the lack of a consistent explanation for the variability in this derived parameter between the different sets of experiments could potentially influence the authors' conclusions. Overall I might argue that caution should be applied to quantitative conclusions based on the technique described until a more complete understanding of the responses is available. Related, does 'r' really represent a consumption ratio? I think it likely represents the equivalence ratio between the influence of added N and P on APA rates within each individual system studied, but it isn't immediately obvious to me that this would be the same as the overall nutrient consumption ratio, in particular due to the potential for variable turnover rates of different pools.

Figure 1: this was useful in helping to understand the technique, however a similar schematic of the opposite case (i.e. where No is negative) would be useful.

Figure 2: I found this figure difficult to interpret on first sight. After using the authors' code to analyse their data I eventually got the hang of what was being presented, however I wonder whether there might be a more simple graphical way to display the data. e.g. could a similar plot to figure 1 be produced with the responses to individual values within experiments contoured/coloured rather than presented as a surface? This may aid a reader in picking out the N = No + rP line and associated intercepts etc. Maybe such a presentation could be used in addition to the format in Figure 2, as the latter admittedly does have the benefit of displaying the magnitude of residuals. Overall I would encourage the authors to consider a wider presentation of the data, as the manuscript is currently short and hence there is ample space available. Related, a key strength of

the data appears to be the repeatability of results, so I would suggest presentation of some more individual experiments would be useful to the reader.

Finally, running the authors MATLAB codes on the data provided I found some discrepancies with the information provided in table 1. I haven't checked all the experiments, but, for example, for the 'M7' experiment for Tvärminne I get:

General model: myfit(x,y) = A/(1+exp(s*(1/sqrt(1+r^2))*(r*x+No-y)))-B Coefficients (with 95% confidence bounds): A = 8.027 (7.681, 8.373) s = 20.67 (19.18, 22.15) r = 1.616 (1.555, 1.677) No = 0.6882 (0.676, 0.7004) B = 0.1094 (0.06078, 0.158)

Thus the estimate for No appears different (albeit only minor) to that stated in the table, as does the confidence interval for s?

Similarly, for 'Fjord' I get:

General model: myfit(x,y) = A/(1+exp(s*(1/sqrt(1+r^2))*(r*x+No-y)))-B Coefficients (with 95% confidence bounds): A = 7.41 (7.156, 7.664) s = 37.87 (34.98, 40.76) r = 2.482 (2.39, 2.573) No = 0.6771 (0.6676, 0.6867) B = 0.2197 (0.1826, 0.2569)

Which again seems different to info in Table 1?

---

## Author Comment (AC1) · 24 Oct 2016

The essential point of the mathematics is to find a surface that fits the data and use this as an objective tool to draw the line separating the P- and N-limited parts of the experimental matrix of N,P-additions. The important argument that this works seems to us that to be that the residuals are relatively small, and with no apparent systematic pattern. The fitting itself is done usingstandard procedures from Matlab, not derived by us.

---

## Author Comment (AC2) · 24 Oct 2016

Authors' response to comments of the Referee 2 uploaded on September 27, 2016. for the manuscript BG-2016-313, Hrustić et al. "Exploring the distance between nitrogen and phosphorus limitation in mesotrophic surface waters using a sensitive bioassay"

General response from authors: We want to thank the referee for his obviously careful reading and helpful comments that we believe has helped to make the text easier accessible. Several of the referee's comments refer to the underlying mechanisms for the response observed and which consequences this has for the interpretation. We agree that this is a difficult, but interesting issue, and feel that the changes made to separate N+ and P+ (see below) from N* and P* was useful. We hope that this

article, with a main scope of documenting the type of responses found, creates the foundation we see as nescessary for the more complicated task of unraveling in detail the mechanisms leading to this type of response.

Referee's comment: "Both conceptually and in terms of practical calculation the estimates of 'N*' and 'P*'derived from the experimental protocol described differ from the typical usage, which has in the past been based on a strict definition (e.g. N* = Nitrate – 16 x Phosphate). In order to avoid confusion within the literature I would prefer the use of different terminology (e.g. maybe 'apparent excess N or P') or, at the very least, a strict definition of the terms used".

Authors' response: We agree that one should distinguish our Âńapparent excess nutrientsÂż as based on bioassays, from the previous definitions of N* and P* calculated from nitrate and phosphate. While N*,P* represent the excess when the system has had a long time in darkness to convert N and P to nitrate and phosphate (combined with the assumption that they will be used in a 16:1 molar ratio), our method is based on detecting when the system state is so that organisms cannot satisfy their need for P, relative to the present availability of N. There is thus a difference in time-scale involved and the numbers therefore do not need to be (and likely are not) identical. We do, however, wish to keep the analogy between the two concepts present in the readers' mind and therefore now introduce the symbols N+ and P+ for what the organisms ÂńseeÂż according to the bioassay. We have modified the introduction to accomodate the definition of N+ and P+.

We absolutely feel that the questions of which mechanisms that make N+,P+ different from N*,P* and on what time scales these mechanisms work, are both important and interesting. The scope of the work presented was, however, the basic question of whether the method would give interesting and reproducible results in environments different from the warm, oligotrophic Eastern Mediterranen (the only environment where it to our knowledge has been used). We feel that publishing the confirmation that this is the case is an important and necessary basis for more elaborate and detailed future studies; by us or or by others that may find the use of this bioassay an interesting option.

Referee's comments: "In particular, at present I remain unsure how and whether the resulting values of derived 'N*' and 'P*' might depend on system characteristics, e.g. including the biological characteristics of the system. Indeed, given the time dependence of the results, it seems likely that derived values are a complex and variable system property. Additionally, I would like to see the authors' thoughts on whether the variable sizes and turnover times of bioavailable dissolved organic N and P pools might influence their results? Additionally, I was unclear how the presence and turnover of natural DOP pools, both AP-hydrolysable and otherwise might be influencing the results?

Authors' response: Since Ânŵhat the organisms seeÂż clearly depends on system state, N+ and P+ are also expected to do so. Among the theoretical possibilities, a phytoplankton community using P:N in a high ratio (both bacteria diatoms probably do) will give another response surface than a bloom using P:N in a lower ratio (flagellates?). Also, if the system successively stores N into DON or into N-rich particulate detritus, the organisms presumably ÂńseeÂż a more and more N-deficient environment. The drift observed in the response surface may thus be explained using the standard assumption that P is remineralized faster than N: The relative surplus of P over N would then increase over time as more N is immobilized in detrital N-pools (including DON). Community composition and trophic successions in the incubation tubes may thus influence the result. This is probably an unavoidable feature of most (all?) bioassays based on incubation: You get the answer for the communities developing in your incubation. This community is obviously selected from the the organisms present in the initial water sample, but likely to be a subset. Our suggested approach is the back-extrapolation to time 0 as explained in the manuscript.

The production of APA should theoretically increase the pool of "bioavailable P" if there is DOP present that is hydrolyzable over the time scale of incubation (days). DOP

therefore should affect the intercept (N0). Because of the feedback effect where "more APA makes more P available", hydrolyzable DOP could also affect the sigmoidity parameter s. It is less obvious that it will affect the slope r. We have wondered whether we could construct a scenario where this feedback mechanism could explain the seemingly low r-values in N-limited systems, but have not been successful in this. DOP that competes with other substrate may also lower the "ceiling" (the parameter A) of the response surface, but this would not affect the interpretation.

Referee's comment: Related to the above, it would be useful to have some additional information on the systems studied, e.g. including inorganic and organic N and P concentrations and potentially some characteristics of the planktonic community?

Authors' response: Our main reason for choosing the two environments was the general expectation of N versus P limitation in the Baltic and in the Norwegian fjord environment, respectively. The response is a system-level response and, as discussed above, a deeper analysis of what happened in the assay would need a lot of information both on chemistry and communiy composition in the system. This was one argument for doing the assays on samples in mesocosms where there would be a huge independent effort on characterizing different ecosystem aspects. For the Tvärminne experiments, interested readers can find a lot of this in doi:10.5194/bg-12-6181-2015 (Paul et al. 2015) and doi:10.5194/bg-13-3035-2016 (Nausch et al. 2016) in the reference list. For the Espegrend mesocosm experiment publication is in preparation, but not yet published. Again, the scope of this work was to establish the type of response found with this method, thereby opening for consequtive work by both ourselves and others describing the set of underlying mechanisms.

Referee's comment: Given the focus of the paper, I would like to see some more details on the APA measurements. In particular what substrate was used (MUFF-P?) and at what concentration.

Authors' response: We have included a slightly more detailed description in th (including the use of 3-O-methylfluorescein-PO4 at final concentration 0.1 $\mu$mol L-1). Modifications of the original method of Perry (1972) were mainly in volume adjustments to match the characteristics of the plate reader in Bergen, while for the Tvärminne all the details of measuring APA were indentical to Perry (1972).

Referee's comment: Also how long were the APA incubations and were multiple timepoints measured within the incubation? Concentrations of substrate are potentially particularly important and I would be interested in the authors' opinion on whether it would matter if the substrate concentration was saturating or not for the derived response surface?

Authors' response: In Bergen, the Perkin-Elmer plate reader was programmed to read each well 15 times within 70 minutes (repetition interval 5 minutes). In Tvärminne, fluorescence on an initial sample was measured after 0, 10, 30 and 60 minutes of incubation with substrate and APA calculated as the slope obtained by linear regression. Based on this, the slope for reported samples are based on a single incubation time ( 30 minutes).

In another western Norwegian fjord, we have estimate the half-saturation constant (K + Sn) for APA with this method to be ca. 300 nM (Thingstad et al. 1993). Our substrate concentration was therefore probably not saturating. A higher substrate concentration is therefore likely to increase the parameter A "lifting the roofÂż of the response surface. Since this parameter is not used in the evaluation of the N:P-kinetics, this would not in itself affect the interpretation of the results. If the concentration effect is not proportional throughout the transition zone between P and N limitation, however, other parameters (No, r, s) could be effected. This has not been investigated. A too low substrate concentration would create a risk for substrate depletion during incubation. With our measurement protocol (above), this should have been detected as a decrease in rate during incubation with substrate and is therefore not likely to have been the case.

Referee's comment: Some of the measured APA values are negative, what does this

represent? I assume some form of blank correction within the fluorometric measurements? Please be more specific on methods.

Authors' response: Some measurements were close to, equal to, and even below the blanks (triplicates for each incubation), whilst the presented measurements (Fig. 2A; Fig. 2B; Supplement) are corrected for blanks. Samples in the N-limited region with no or neglible APA therefore get negative values. This will affect the parameter B describing the "floor" of the response surface. B is not used in the interpretation of N:P relationships.

Referee's comments and suggestions how to rewrite the manuscript (authors' responses are below each comment)

Abstract:

Referee: Line 19: I suggest '. . .primary limiting nutrient. . .'

Authors: We have corrected the Lines 18–22 into: This assay not only provides information on which element (N or P) is the primary limiting nutrient, but also gives a quantitative estimate for the excess of the secondary limiting element (P+ or N+, respectively), as well as the ratio of balanced consumption of added N and P over short time scales (days).

Section 1:

Referee: Line 16: suggest '. . .fixers potentially having a competitive. . .' i.e. the evidence in the cited study is largely on the basis of hypothetical modelling

Authors: "potentially" is included in the sentence.

Referee: Line 22: define N* and P* (e.g. N* = Nitrate – 16 x Phosphate) on first use (also note above, I would prefer use of different terminology for derived apparent excesses of N and P)

Authors: Done

Referee: Line 44: It was not immediately clear to me that the assays described can be used to quantitatively asses the excess of overall bioavailable N or P in a system and, despite spending some time with the authors' code and data (which are very usefully provided within the supplement), I still remain to be entirely convinced. Firstly, there is clearly an assumption that the organisms in the community react to P limitation through increasing APA (see abstract of Thingstad and Mantoura 2005). While this may be the case, it remains an assumption and should probably be explicitly stated as such. I think there is also an underlying assumption that the community level APA response is directly proportional to the overall availability of all forms of bioavailable N and P within the system.

Authors: It is a standard assumption based on the literature that the organisms within the microbial community react to P-limitation by inducing production of the alkaline phosphatase enzyme. A standard measurement of APA (without prior incubation with N and P) is thus a convenient and frequently used way to determine whether the organisms "feel" P-limited (have de-repressed the Pho-operon). We have re-read our introduction and feel that the point that microbes react to P-limitation by inducing AP is relatively clear. Our assay thus basically should measure when organisms have consumed the available forms of P in the incubation tubes, but still have N available to produce the enzyme. As discussed previously in the our reply, the result therefore should depends on the structure of the community: E.g. a community that binds a lot of P relative to N will give a larger P-limited region in the N:P matrix. We are not sure that linearity in the response to bioavailable N and P is an assumption. With strict linearity, the transition between the N and P limited regions should probably be linear and not sigmoid (?) The sigmoidal shape may originate from flexibility in the biological response (variability in the response of individuals or species or functional groups). Our sigmoid response surface was originally choosen from the analogy to LC50 analysis in toxicology (See Thingstad and Mantoura) to fit the observed response, not from a mechanistic model of AP-production.

[Figure]

Referee: The authors can correct me if this is wrong, however, I think that the equation on line 2, page 4 suggests there is this assumption of a linearity of response? Overall however I will admit that I struggled to follow some of the authors' arguments, particularly on the first read through. I would therefore suggest that the authors could provide a fuller treatment and explanation for their analysis method for the data presented.

Authors: The line does not represent the linearity in the particular response of APA in a certain microcosm. It represents an assumed linearity in the consumption of N and P by the community. The validity of this assumption is confirmed by our results. If N and P were used in a different ratio in the upper right corner of the addition matrix (high additions) than in the lower left corner (low additions) this would have been visible as a systematic pattern in the residuals shown in Fig.2.

Referee: Related, it would be good to fully separate out the equations, number them and explicitly refer to them as/when required. Some additional simple graphical schematic plots and further graphical presentation of the extensive data set might also aid accessibility.

Authors: We usually agree in the strategy to take equations out of the text and place them in a separate box/table. In this case, however, there is only one basic equation: the fitting function. We feel that the box/table solution therefore is not suitable here. We have tried to make this a bit more readable by the minor modification of separating this equation out as a separate paragraph. The two figures in Fig.2A and 2B represent the N- and P-limited situations. Similar plots of the other data seem to us to give minor additional information only. We do not see other graphs that contain enough new information to warrant the space required.

Section 2:

Referee: Line 4: Rephrase. Please provide some background to the experiment. At this stage the experimental mesocosms have not been described/introduced.

Authors: We have added an introductory paragraph about the mesocosms and their main purpose (acidification effects). Since we found no significant acidifaction effects on our assays, we haave chosen not to go into details which the interested readers can find in the reference cited.

Referee: Line 30 (section 2.3): please provide information on both the incubation time for APA, substrate used, concentration etc. (see above). Authors: Done

Referee: Page 6, lines 1-12: given that the value of r has a direct impact on the calculations of any excess bioavailable N or P within the system, I was left wondering how the lack of a consistent explanation for the variability in this derived parameter between the different sets of experiments could potentially influence the authors' conclusions.

Authors: The variation and the seemingly consistent difference in r between N and P limited environments is interesting. We again feel that the finding itself is properly reported and discussed. To determine the underlying mechanisms and the consequences for our understanding of the system assayed would require research beyond the scope of this work.

Referee: Overall I might argue that caution should be applied to quantitative conclusions based on the technique described until a more complete understanding of the responses is available. Related, does 'r' really represent a consumption ratio? I think it likely represents the equivalence ratio between the influence of added N and P on APA rates within each individual system studied, but it isn't immediately obvious to me that this would be the same as the overall nutrient consumption ratio, in particular due to the potential for variable turnover rates of different pools.

Authors: The remineralization issue is discussed above. As a response to this comment we have tried to increase the precision of our language by changing "consumption" to "net consumption" which is what determines the relative sizes of the pools of free N and P.

[Figure]

Referee: Figure 1: this was useful in helping to understand the technique, however a similar schematic of the opposite case (i.e. where No is negative) would be useful.

Authors: This is explained and Fig.2 contains examples of the two cases. We feel that the additional explanation contained in a Fig. 1B with the line shiftet to give negative intercept with the y-axis would be too marginal.

Referee: Figure 2: I found this figure difficult to interpret on first sight. After using the authors' code to analyse their data I eventually got the hang of what was being presented, however I wonder whether there might be a more simple graphical way to display the data. e.g. could a similar plot to figure 1 be produced with the responses to individual values within experiments contoured/coloured rather than presented as a surface? This may aid a reader in picking out the N = No + rP line and associated intercepts etc. Maybe such a presentation could be used in addition to the format in Figure 2, as the latter admittedly does have the benefit of displaying the magnitude of residuals. Overall I would encourage the authors to consider a wider presentation of the data, as the manuscript is currently short and hence there is ample space available. Related, a key strength of the data appears to be the repeatability of results, so I would suggest presentation of some more individual experiments would be useful to the reader.

Authors: We chose this representation exactly from the reason identified by the referee: the possibilty to represent the residuals. We see these as important for demonstrating that the fitting function chosen is suitable (in terms of representing the observed response).

Referee: Finally, running the authors MATLAB codes on the data provided I found some discrepancies with the information provided in table 1. I haven't checked all the experiments, but, for example, for the 'M7' experiment for Tvärminne I get: General model: myfit(x,y) = A/(1+exp(s*(1/sqrt(1+rËĘ2))*(r*x+No-y)))-B Coefficients (with 95= 1.616 (1.555, 1.677) No = 0.6882 (0.676, 0.7004) B = 0.1094 (0.06078, 0.158) Thus the

estimate for No appears different (albeit only minor) to that stated in the table, as does the confidence interval for s? Similarly, for 'Fjord' I get: General model: myfit(x,y) = A/(1+exp(s*(1/sqrt(1+rËEȩ2))*(r*x+No-y)))-B Coefficients (with 95= 2.482 (2.39, 2.573) No = 0.6771 (0.6676, 0.6867) B = 0.2197 (0.1826, 0.2569) Which again seems different to info in Table 1?

A = 16.05 (13.34, 18.76) s = 29.19 (23.11, 35.27) r = 2.52 (2.214, 2.825) No = 0.7897 (0.7473, 0.8321) B = 0.2284 (0.08769, 0.369)

Authors: Re-running our codes we find the values given in Table 1. To prevent a possible mixup of file-titles we will re-load the files in SI if the article is accepted.

References used in the answers to the comments of the Reviewer 2

Nausch, M. (1998) Alkaline phosphatase activities and the relationship to inorganic phosphate in the Pomeranian Bight (southern Baltic Sea). Aquatic Microbial Ecology 16(1), 87–94. Paul, A. J., Bach, L. T., Schulz, K. G., Boxhammer, T., Czerny, J., Achterberg, E. P., Hellemann, D., Trense, Y., Nausch, M., Sswat, M. Riebesell, U. Effect of elevated CO2 on organic matter pools and fluxes in a summer Baltic Sea plankton community. Biogeosciences 12, 6181-6203, doi:10.5194/bg-12-6181-2015 (2015). Thingstad, T. F., Skjoldal, E. F. Bohne, R. A. Phosphorus cycling and algal-bacterial competition in Sandsfjord, western Norway. Mar.Ecol.Prog.Ser. 99, 239-259 (1993).

---

## Author Comment (AC3) · 24 Oct 2016

We want to thank M. Krom for his positive evaluation and comments. For his comment on the mathematics, see reply to first posting of the referee comment.

---

## Author Response (AR1)

**Letter to Editor and response to referee comments for revised version of BG-2016-313,**
**Hrustić et al. „Exploring the distance between nitrogen and phosphorus limitation in**
**mesotrophic surface waters using a sensitive bioassay"**
**General response from authors**: We want to thank referee#1 for his positive
evaluation and referee #2 for her/his obviously careful reading and helpful comments.
We have given special attention to referee #2's concerns on accessability and readability
and hope that our revisions have improved the text in this respect. Several of the
referee's comments refer to the underlying mechanisms for the response observed and
which consequences this has for the interpretation. We absolutely agree that this is
interesting and important and we have expanded both the introduction and the
discussion on this issue. We also agree with the referee that this bioassay gives results
that are likely to be different from N* and P*, and therefore introduced the symbols $N^+$
and $P^+$ to distinguish bioassayed surplus nutrients from the traditional, chemically
determined N* and P*.
Since our measurements of the processes in our incubation tubes were restricted to the
APA determinations, any discussion on processes, populations etc. remain hypothetical.
We have therefore tried to find a balance between the need to point out possible
processes influencing our results, and the danger of long arguments that may be appear
too speculative to some readers.
**Referee's comment:** „Both conceptually and in terms of practical calculation the estimates
of 'N*' and 'P*'derived from the experimental protocol described differ from the typical
usage, which has in the past been based on a strict definition (e.g. N* = Nitrate – 16 x
Phosphate). In order to avoid confusion within the literature I would prefer the use of
different terminology (e.g. maybe 'apparent excess N or P') or, at the very least, a strict
definition of the terms used".
**Authors' response:** We agree that one should distinguish our «apparent excess nutrients» as
based on bioassays, from the previous definitions of N* calculated from nitrate and
phosphate. We do, however, wish to keep the analogy between the two concepts present in
the readers' mind and therefore now introduce the symbols $N^+$ and $P^+$ for what the
organisms «see» according to the bioassay. We have modified the introduction to
accomodate the definition of $N^+$ and $P^+$.
**Referee's comments:** „In particular, at present I remain unsure how and whether the
resulting values of derived 'N*' and 'P*' might depend on system characteristics, e.g.
including the biological characteristics of the system.
Indeed, given the time dependence of the results, it seems likely that derived values are a
complex and variable system property. Additionally, I would like to see the authors' thoughts
on whether the variable sizes and turnover times of bioavailable dissolved organic N and P
pools might influence their results?
Additionally, I was unclear how the presence and turnover of natural DOP pools, both AP-
hydrolysable and otherwise might be influencing the results?
**Authors' response:** We have expanded both the introduction and the discussion to make the
reader aware of what we think are the most likely processes influencing our results. We have tried to balance this against the danger of too extensive, experimentally unsupported and speculative discussions.

**Referee's comment:** Related to the above, it would be useful to have some additional information on the systems studied, e.g. including inorganic and organic N and P concentrations and potentially some characteristics of the planktonic community?

**Authors' response:** Our main reason for choosing the two environments was the general expectation of N versus P limitation in the Baltic and in the Norwegian fjord environment, respectively. The response is a system-level response and, as discussed above, a deeper analysis of what happened in the assay would need a lot of information both on chemistry and community composition in the system. This was one argument for doing the assays on samples in mesocosms where there would be huge independent efforts on characterizing different ecosystem aspects. For the Tvärminne experiments, interested readers can find a lot of this in doi:10.5194/bg-12-6181-2015 (Paul et al., 2015) and doi:10.5194/bg-13-3035-2016 (Nausch et al., 2016), both in the reference list. The Espegrend mesocosm experiment is not yet published.

**Referee's comment:** Given the focus of the paper, I would like to see some more details on the APA measurements. In particular what substrate was used (MUFF-P?) and at what concentration. Also how long were the APA incubations and were multiple timepoints measured within the incubation? Concentrations of substrate are potentially particularly important and I would be interested in the authors' opinion on whether it would matter if the substrate concentration was saturating or not for the derived response surface?

**Authors' response:** We have included a more detailed description in M&M, the use of 3-o-methylfluorescein-$PO_4$ at final concentration 0.1 µmol L$^{-1}$. Modifications of the original method of Perry (1972) were mainly in volume adjustments to match the characteristics of the plate reader in Bergen, while for the Tvärminne all the details of measuring APA were indentical to Perry (1972). In Bergen, the Perkin-Elmer plate reader was programmed to read each well 15 times within 70 minutes (repetition interval 5 minutes).
In Tvärminne, fluorescence on an initial sample was measured after 0, 10, 30 and 60 minutes of incubation with substrate and APA calculated as the slope obtained by linear regression. Based on this, the slope for reported samples are based on a single incubation time (30 minutes).
In another western Norwegian fjord, we have estimated the half-saturation constant (K+Sn) for APA with this substrate to be ca. 300 nM (Thingstad et al., 1993; see the manuscript). Our substrate concentration was therefore probably not saturating. A higher substrate concentration is therefore likely to increase the parameter A „lifting the roof" of the response surface. Since this parameter is not used in the evaluation of the N:P-kinetics, this would not in itself affect the interpretation of the results. If the concentration effect is not proportional throughout the transition zone between P and N limitation, however, other parameters ($N_0$, r, s) could in principle be affected. This has not been investigated.
A too low substrate concentration would create a risk for substrate depletion during incubation. With our measurement protocol (above), this should have been detected as a decrease in rate during incubation with substrate and is therefore not likely to have been the case.

**Referee's comment:** Some of the measured APA values are negative, what does this
represent? I assume some form of blank correction within the fluorometric measurements?
Please be more specific on methods.

**Authors' response:** Some measurements were close to, equal to, and even below the blanks
(triplicates for each incubation), whilst the presented measurements (Fig. 2A; Fig. 2B;
Supplement) are corrected for blanks. Samples in the N-limited region with no or neglible
APA therefore get negative values. This will affect the parameter B describing the „floor" of
the response surface. B is not used in the interpretation of N:P relationships.

**Referee's comments and suggestions how to rewrite the manuscript**
**(authors' responses are below each comment)**

**Abstract:**

**Referee:** Line 19: I suggest '. . .primary limiting nutrient. . .'

**Authors:** We have corrected the Lines 18–22 into: This assay not only provides information
on which element (N or P) is the primary limiting nutrient, but also gives a quantitative
estimate for the excess of the secondary limiting element ($P^+$ or $N^+$, respectively), as well as
the ratio of balanced consumption of added N and P over short time scales (days).

**Section 1:**

**Referee:** Line 16: suggest '. . .fixers potentially having a competitive. . .' i.e. the evidence in
the cited study is largely on the basis of hypothetical modelling

**Authors:** „potentially" is included in the sentence.

**Referee:** Line 22: define $N^*$ and $P^*$ (e.g. $N^*$ = Nitrate – 16 x Phosphate) on first use (also
note
above, I would prefer use of different terminology for derived apparent excesses of N and P)

**Authors:** Done

**Referee:** Line 44: It was not immediately clear to me that the assays described can be used
to quantitatively asses the excess of overall bioavailable N or P in a system and, despite
spending some time with the authors' code and data (which are very usefully provided
within the supplement), I still remain to be entirely convinced. Firstly, there is clearly an
assumption that the organisms in the community react to P limitation through increasing
APA (see abstract of Thingstad and Mantoura 2005). While this may be the case, it
remains an assumption and should probably be explicitly stated as such. I think there
is also an underlying assumption that the community level APA response is directly
proportional to the overall availability of all forms of bioavailable N and P within the
system.

**Authors:** It is a standard assumption based on the literature that the organisms within the microbial community react to P-limitation by inducing production of the alkaline phosphatase enzyme. A standard measurement of APA (without prior incubation with N and P) is thus a convenient and frequently used way to determine whether the organisms „feel" P-limited (have de-repressed the Pho-operon). We have re-read and slightly adjusted the wording of our introduction and feel that the point that microbes react to P-limitation by inducing AP synthesis now should be relatively clear. Our assay thus basically should measure when organisms have consumed the available forms of P in the incubation tubes, but still have N available to produce the enzyme. The result therefore should depend on the structure of the community: E.g. a community that binds a lot of P relative to N will give a larger P-limited region in the N:P matrix. We are not sure that linearity in the response to bioavailable N and P is an assumption. With strict linearity, the transition between the N and P limited regions should probably be linear and not sigmoid (?) The sigmoidal shape may originate from flexibility in the biological response (variability in the response of individuals or species or functional groups). Our sigmoid response surface was originally choosen from the analogy to $LC_{50}$ analysis in toxicology (see Thingstad and Mantoura, 2005) to fit the observed response, not from a mechanistic model of AP-production.

**Referee:** The authors can correct me if this is wrong, however, I think that the equation on line 2, page 4 suggests there is this assumption of a linearity of response? Overall however I will admit that I struggled to follow some of the authors' arguments, particularly on the first read through. I would therefore suggest that the authors could provide a fuller treatment and explanation for their analysis method for the data presented.

**Authors:** The line does not represent linearity in the particular response of APA in a certain microcosm (i.e. falcon tube). It represents an assumed linearity in the consumption of N and P by the community. The validity of this assumption is confirmed by our results. If N and P were used in a different ratio in the upper right corner of the addition matrix (high additions) than in the lower left corner (low additions) this should have been visible as a systematic pattern in the residuals shown in Fig.2.

**Referee:** Related, it would be good to fully separate out the equations, number them and explicitly refer to them as/when required. Some additional simple graphical schematic plots and further graphical presentation of the extensive data set might also aid accessibility.

**Authors:** We usually agree in the strategy to take equations out of the text and place them in a separate box/table. In this case, however, there are only three equations, and they are only used in this already technical section. We feel that the box/table solution therefore is not optimal here.
We have instead tried to make the text a bit more readable by separating the three equations out as separate paragraphs and numbering them, allowing later reference. We have also added a bit more text to hopefully help the reader to what is probably a more unconventional than a particularly difficult algorithm.

We have also re-coloured the fitted response-surfaces in Fig. 2, giving them a gradient in colour. Hopefully this helps the intuitive impression of the shape of these surfaces.

The two figures in Fig. 2A and 2B represent the N- and P-limited situations, respectively. Similar plots of the other data seem to us to give minor additional information only. We have not been able to come up with other graphical representations that provide fundamental new insights into the data.

**Section 2:**

**Referee:** Line 4: Rephrase. Please provide some background to the experiment. At this stage the experimental mesocosms have not been described/introduced.

**Authors:** We have added an introductory line about the mesocosms and their main purpose (acidification effects). Since we found no significant acidifaction effects on our assays, we haave chosen not to go into details. For the Tvärminne experiment, the interested reader can find this in the references cited. Documentation of the Espegrend mesocosm experiment is so far in preparation and will be available to future readers.

**Referee:** Line 30 (section 2.3): please provide information on both the incubation time for APA, substrate used, concentration etc. (see above).

**Authors:** Done

**Referee:** Page 6, lines 1-12: given that the value of r has a direct impact on the calculations of any excess bioavailable N or P within the system, I was left wondering how the lack of a consistent explanation for the variability in this derived parameter between the different sets of experiments could potentially influence the authors' conclusions.

**Authors:** The variation and the seemingly consistent difference in r between N and P limited environments is interesting. We again feel that the finding itself is properly reported and discussed. To determine the underlying mechanisms and the consequences for our understanding of the system assayed would require research beyond the scope of the present work.

**Referee:** Overall I might argue that caution should be applied to quantitative conclusions based on the technique described until a more complete understanding of the responses is available. Related, does 'r' really represent a consumption ratio? I think it likely represents the equivalence ratio between the influence of added N and P on APA rates within each individual system studied, but it isn't immediately obvious to me that this would be the same as the overall nutrient consumption ratio, in particular due to the potential for variable turnover rates of different pools.

**Authors:** As a response to this comment we have tried to increase the precision of our language by changing „consumption" to „net consumption" which is what determines the relative sizes of the pools of free N and P. We have also added a caution in the final conclusion that the underlying mechanisms need further investigation.

**Referee:** Figure 1: this was useful in helping to understand the technique, however a similar schematic of the opposite case (i.e. where No is negative) would be useful.

**Authors:** This is now explained in the legend of Fig. 1, whilst Fig. 2 contains examples of the two cases. We feel that adding more information (e.g. a line with negative Y-intercept) would clutter the presentation. The alternative of a Fig. 1B with the line shifted to give a negative intercept is indeed possible, but was felt to give marginal new information.

**Referee:** Figure 2: I found this figure difficult to interpret on first sight. After using the authors' code to analyse their data I eventually got the hang of what was being presented, however I wonder whether there might be a more simple graphical way to display the data. e.g. could a similar plot to figure 1 be produced with the responses to individual values within experiments contoured/coloured rather than presented as a surface? This may aid a reader in picking out the N = No + rP line and associated intercepts etc. Maybe such a presentation could be used in addition to the format in Figure 2, as the latter admittedly does have the benefit of displaying the magnitude of residuals. Overall I would encourage the authors to consider a wider presentation of the data, as the manuscript is currently short and hence there is ample space available. Related, a key strength of the data appears to be the repeatability of results, so I would suggest presentation of some more individual experiments would be useful to the reader.

**Authors:** We chose this representation exactly from the reason identified by the referee: the possibilty to represent the residuals. We see these as important for demonstrating that the fitting function chosen is suitable (in terms of representing the observed response). A contour plot would be a series of lines parallel to the $N = N_0 + rP$ line and did not seem very informative to us. We have tried to meet the need for an intuitively easier graphics by changing the previously uniform gray response surface to a colour gradient (blue to red) .

**Referee:** Finally, running the authors MATLAB codes on the data provided I found some discrepancies with the information provided in table 1. I haven't checked all the experiments, but, for example, for the 'M7' experiment for Tvärminne I get:
General model: myfit(x,y) = A/(1+exp(s*(1/sqrt(1+r^2))*(r*x+No-y)))-B Coefficients (with 95% confidence bounds): A = 8.027 (7.681, 8.373) s = 20.67 (19.18, 22.15) r = 1.616 (1.555, 1.677) No = 0.6882 (0.676, 0.7004) B = 0.1094 (0.06078, 0.158)
Thus the estimate for No appears different (albeit only minor) to that stated in the table, as does the confidence interval for s?
Similarly, for 'Fjord' I get:
General model: myfit(x,y) = A/(1+exp(s*(1/sqrt(1+r^2))*(r*x+No-y)))-B Coefficients (with 95% confidence bounds): A = 7.41 (7.156, 7.664) s = 37.87 (34.98, 40.76) r = 2.482 (2.39, 2.573) No = 0.6771 (0.6676, 0.6867) B = 0.2197 (0.1826, 0.2569)
Which again seems different to info in Table 1?

```
  A =     16.05  (13.34, 18.76)
   s =     29.19  (23.11, 35.27)
   r =      2.52  (2.214, 2.825)
  No =     0.7897  (0.7473, 0.8321)
   B =    0.2284  (0.08769, 0.369)
```

**Authors:** Re-running our codes we find the values given in Table 1. To prevent a possible mixup of file-titles we will re-load the files in SI if the article is accepted.

Please find below our manuscript with the track-changes option on, showing the main changes from the original ms.

[revised manuscript text omitted]

---

## Author Response (AR2)

Bergen 16 December 2016

Dear Editor:

Response to the request for minor revision of ms bg-2016-313

We tried several options to include examples of balanced and P-limited situations in Fig.1, without cluttering the already detail-rich figure too much. We ended up with the suggestion below where we use a small extra panel with colors to illustrate the three cases.

We have also shaded the P-limited region in Panel A and we discovered a typo in one of the equations that has now been corrected.

The legend is changed accordingly.

Best regards

T. Frede Thingstad

[Figure]

Figure 1. Illustration of the fitting algorithm used. With APA measured over a 4x7 matrix of combinations in additions of P and N (black dots, Panel A), the objective is to find the line that splits this $P,N$-plane in an upper P-limited region with high APA (shaded, Panel A) and a lower N-limited region with low APA. This is done by least square fitting of the surface $APA_{est} = \frac{A}{1+e^{sZ}} - B$ to the APA-values measured in each grid point. $APA_{est}$ is a sigmoidal function of the perpendicular distance $Z = \frac{1}{\sqrt{1+r^2}}(rP + (N - N_0))$ from the point $P,N$ (marked X in Panel A) to the line. The situation illustrated in Panel A represents an N-limited system with the positive $N$-axis intercept ($N_0$) and excess-P (P[+]) represented by the negative $P$-axis intercept $N_0/r$. Panel B illustrates the separating line for three hypothetical situations characterized by N-limitation (blue), N/P balance (red) and P-limitation (green). All three with r = 16, but with P[+] = 0.625 µM-P, P[+] = N[+] = 0, and N[=] = 10 µM-N for the N-limited, the balanced, and the P-limited situation, respectively.